# Physicochemical, Spectroscopic and Chromatographic Analyses in Combination with Chemometrics for the Discrimination of Four Sweet Cherry Cultivars Grown in Northern Greece

**DOI:** 10.3390/foods8100442

**Published:** 2019-09-26

**Authors:** Spyridon Papapetros, Artemis Louppis, Ioanna Kosma, Stavros Kontakos, Anastasia Badeka, Chara Papastephanou, Michael G. Kontominas

**Affiliations:** 1Laboratory of Food Chemistry, Department of Chemistry, University of Ioannina, 45110 Ioannina, Greece; spirosp32@gmail.com (S.P.); ikosma@cc.uoi.gr (I.K.); abadeka@uoi.gr (A.B.); 2cp Foodlab Ltd., Polifonti, 25, 2047 Strovolos, Nicosia, Cyprus; artemislouppis@gmail.com (A.L.); foodlab@cytanet.com.cy (C.P.); 3Department of Social Administration and Political Science, Democritus University of Thrace, 69100 Komotini, Greece; skon2915dt@gmail.com

**Keywords:** sweet cherries, discrimination, physicochemical quality parameters, chemometrics

## Abstract

A total of 56 sweet cherry samples belonging to four cultivars (Ferrovia, Canada Giant, Lapins, and Germersdorfer) grown in northern Greece were characterized and differentiated according to botanical origin. For the above purpose, the following parameters were determined: conventional quality parameters (titratable acidity (TA), pH, total soluble solids (TSS), total phenolic content (TPC), mechanical properties and sensory evaluation, sugars by High Performance Liquid Chromatography (HPLC), volatile compounds by GC/MS, and minerals by ICP-OES. Statistical treatment of the data was carried out using Multivariate Analysis of Variance (MANOVA) and Linear Discriminant Analysis (LDA). The results showed that the combination of volatile compounds and conventional quality parameters provided a correct classification rate of 84.1%, the combination of minerals and conventional quality parameters 86.4%, and the combination of minerals, conventional quality parameters and sugars provided the highest correct classification rate of 88.6%. When the above four cherry cultivars were combined with previously studied Kordia, Regina, Skeena and Mpakirtzeika cultivars, collected from the same regions during the same seasons, the respective values for the differentiation of all eight cultivars were: 85.5% for the combination of conventional quality parameters, volatiles and minerals; and 91.3% for the combination of conventional quality parameters, volatiles, minerals, and sugars.

## 1. Introduction

Cherries have been cultivated for thousands of years in Europe, as pits have been recovered from cave dwellings dating back to 4000–5000 B.C. Today, cherry cultivation is very popular in Greece [1]. Sweet cherries (*Prunus avium L*.) are widely accepted for their quality characteristics, such as skin color, texture, sugar and organic acid content, and volatile compound composition [2].

Sweet cherries are a natural source of useful ingredients such as phenolic compounds functioning as natural antioxidants, which reduce the risk of degenerative diseases caused by oxidative stress; they are also a source of minerals, sugars, and organic acids [3,4,5].

Sugars are one of the main ingredients of sweet cherries, which, along with organic acids, lead to the unique balance of fruit flavor. The sugar content can be as high as one-quarter of the total weight of the fruit [6]. Of the different sugars normally found in sweet cherries (glucose, sucrose, fructose, maltose, and sorbitol), glucose and fructose account for approximately 90% of the total sugars in the fruit [7].

Sensory quality attributes of cherries include sweetness (sugar content), sourness (organic acid content), fruit weight, skin color, fruit firmness, and particular aroma. Usually aroma, making up a very small portion of fruit weight (0.01%–0.001% fresh weight basis), comprises of a mixture of a large number of volatile and semi-volatile compounds collectively contributing to fruit odor [8,9]. Classes of volatile aroma compounds include aldehydes, alcohols, esters, acids, terpenes, etc. Particular volatile compounds are characteristic of fruit cultivar and can, thus, be used for the differentiation of cherry cultivars.

Based on the above, the main objective of the present study was to characterize four popular cherry cultivars (Ferrovia, Canada Giant, Lapins, and Germersdorfer) grown in Northern Greece and to differentiate them according to cultivar. The Ferrovia cultivar was developed in the area of Puglia, Italy. It is cultivated in many geographical regions in Greece. The fruit is large, heart-shaped with a red, shiny skin, and a firm texture. The fruit flesh is pink in color, juicy, with a strong adherence to the stone. The flavor is of moderate sweetness [10]. The Canada Giant cultivar originates from Summerland, Canada. It gives a large, red, shiny skin, heart-shaped fruit of firm texture. Greece, due to its mild climate, has the comparative advantage of early sweet cherry ripening by 10–15 days as compared to the rest of Europe, with the exception of Turkey. The Lapins cultivar was developed by K.O. Lapin at the Agricultural Research Station in Summerland, British Columbia in Canada. This cherry cultivar gives a large, red, shiny skin, heart-shaped fruit of firm texture. The fruit flesh is very juicy and dark red in color [11]. The Germersdorfer cultivar is of Hungarian origin, grown in all central European countries with different names. It is a firm, crisp, juicy, aromatic, sweet, slightly acidic cherry variety with a bright red fruit color [1].

As of 1992, the European Union has set specific rules defining the status of products labeled as protected destination of origin (PDO), protected geographical indication (PGI), and traditional specially guaranteed (TSG). Such an act spurs the production of unique foods on the basis of their territorial, compositional and sensory traits, as well as their preparation methods. Such foodstuff is of higher value, both in domestic and international markets [12,13].

Techniques used for the authentication of foodstuff include HPLC, ICP-OES, GC/MS, IRMS, Electronic Nose, NMR, DNA analysis, etc. [14,15,16,17].

The present study comprises of the second half of the work recently published [17] by our group focusing on different four popular sweet cherry cultivars: Ferrovia, Canada Giant, Lapins, and Germersdorfer, also grown in northern Greece. Therefore, a second objective, considered a challenge given the large number of cherry cultivars studied all together, was to attempt to differentiate all eight cultivars, those studied in our previous wοrk and those studied in the present work.

## 2. Materials and Methods

### 2.1. Samples

Fifty-six sweet cherry samples were collected from northern Greece (Edessa and neighbouring Kozani region) as follows: 19 (10 + 9, for each of two harvesting periods) samples of the Ferrovia cultivar, 12 (6 + 6) samples of the Canada Giant cultivar, 13 (7 + 6) samples of the Lapins cultivar, and 12 (6 + 6) samples of the Germersdorfer cultivar. Samples were collected during the period of 20 May–30 June for two consecutive years (2015–2016) at the stage of full ripeness, placed in rectangular plastic cups (500 g per cup), and transferred to the laboratory in insulated styrofoam boxes within four hours after collection.

### 2.2. Determination of Conventional Quality Parameters

Conventional quality parameters total soluble solids (TSS), pH, titratable acidity (TA), and mechanical properties were determined according to the methods described by Papapetros et al. [17]. TSS was measured by refractometry and expressed as Brix. TA was measured volumetrically and expressed as g of Malic acid equivalents (MAE) per 100 g Fresh Weight (FW).

Mechanical properties (force/load and penetration) were determined by dynamometry and expressed as (N) and (mm), respectively. TPC was determined according to Vavoura et al. [11] by adding the methanolic cherry extract to the Folin-Ciocalteu reagent and upon completion of the reaction, measurement of the absorbance at λ = 725 nm. Aqueous solutions of Gallic acid were used for quantification of TPC. Results were expressed as mg of Gallic acid equivalents/ 100 g FW.

Sensory evaluation was carried out according to Vavoura et al. [11] using a 51-member panel (acceptability test). Panelists evaluated color, texture and flavor on a 5-point hedonic scale with 5 corresponding to the most liked sample and 1 corresponding to the least liked sample. A score of 3 was taken as the lower limit of acceptability.

### 2.3. Determination of Sugars Using HPLC-RID

The main sugars in cherries are glucose and fructose. Sugar analysis, including method analytical characteristics, are described by Papapetros et al. [17]. All measurements were carried out in triplicate.

### 2.4. Identification and Semi-Quantification of Volatile Compounds Using Solid Phase Micro-Extraction in Combination with Gas Chromatography/Mass Spectrometry (SPME-GC/MS)

The analysis of volatile compounds was carried out using the method by Vavoura et al. [11].

### 2.5. Determination of Minerals Using Inductively Coupled Plasma Optical Emission Spectrometry (ICP-OES)

Elemental analysis was carried out according to the method described by Papapetros et al. [17].

### 2.6. Statistical Analysis

All measurements were carried out three-fold (*n* = 3) with the exception of mechanical properties carried out five-fold (*n* = 5). Analytical data were treated using the SPSS 23.0 Statistics software (IBM, Armonk, NY, USA) as described by Papapetros et al. [17].

## 3. Results and Discussion

### 3.1. Determination of Conventional Quality Parameters

The results for the conventional quality parameters are presented in Table 1. The values for cherry pH are quite similar, ranging from 3.98 ± 0.20 to 4.14 ± 0.09 for the Germersdorfer and the Canada Giant cultivars, respectively. Similar pH values were reported by Vursavuş et al. [18] for the three cheery cultivars: Van, Noir De Guben, and 0-900 Ziraat grown in Turkey (3.82, 4.20, and 4.10, respectively). Vavoura et al. [11] analysed cherries from four cultivars (Canada Giant, Lapins, Ferrovia, and Skeena) and recorded pH values of 3.91 ± 0.00, 3.96 ± 0.00, 3.81 ± 0.00 and 3.91 ± 0.00, respectively. Papapetros et al., [17] reported similar pH values for cherry cultivars: Kordia, Regina, Mpakirtzeika, and Skeena; with Regina recording the highest pH value (4.69). Tural and Koca [19] studied Cornelia cherry fruits and recorded lower pH values than those of the present study, ranging from 3.11 to 3.53. Likewise, Chockchaisawasdee et al. [20] observed variations in pH values among the 56 cherry cultivars analysed, ranging from 3.5 for the Van cultivar to 4.8 for the Minnulara cultivar.

TSS values of the tested cultivars ranged from 12.23 ± 1.99 Brix for the Lapins cultivar to 14.06 ± 2.27 Brix for the Ferrovia cultivar. Vursavuş et al. [18] recorded TSS values of 14.20% for the Van, 14.00% for the Noir De Guben, and 14.40% for the 0-900 Ziraat cultivar. On the other hand, Vavoura et al. [11] recorded higher TSS values for the cultivars: Canada Giant, Lapins, and Ferrovia (14.50 ± 0.05, 13.00 ± 0.03, and 16.00 ± 0.06 Brix, respectively). Papapetros et al. [17] reported similar TSS values to those of the present study for cherry cultivars: Kordia, Regina, Mpakirtzeika, and Skeena, ranging between 12.8 and 14.6 Brix. Serradilla et al. [7] observed significant differences in TSS values, ranging from 13.97 Brix for the Sweetheart to 23.20 Brix for Pico Colorado. Likewise, Hayaloglu and Demir [21] investigated 12 cherry cultivars and recorded TSS values ranging from 13.71 ± 0.47 Brix for the Durona di Cesena to 19.55 ± 0.29 Brix for the Bing cherry cultivar. Tural and Koca [19] studied the Cornelia cherry cultivar and reported TSS values of 76.80–154.00 g/kg. For the Lapins cultivar, Hayaloglu and Demir [21] recorded a TSS value of 17.52 Brix, much higher than the Lapins TSS values of the present study.

TA was expressed as MAE per 100 g FW. TA values ranged from 0.27 ± 0.04 g MAE/100 g FW for the Germersdorfer cultivar to 0.35 ± 0.04 g MA/100 g FW for the Canada Giant. Lower TA values have been recorded by Vavoura et al. [11] for the four cultivars, Canada Giant, Lapins, Ferrovia, and Skeena (0.28 ± 0.01, 0.20 ± 0.00, 0.27 ± 0.00, and 0.21 ± 0.01 g MA/100g FW, respectively). Papapetros et al. [17] reported similar TA values to those of the present study for cherry cultivars: Kordia, Regina, Mpakirtzeika, and Skeena, ranging between 0.19 and 0.39 g MAE/100 g. Esti et al. [22] recorded higher TA values for the Ferrovia cultivar (0.81 ± 0.05 g/100 g) than those of the present study. Vursavuş et al. [18] observed differences in TA values of TA, ranging from 4.75 ± 0.01 g/L for the Noir De Gubento to 7.08 ± 0.01 g/L for the Van cultivar. According to Hayaloglu and Demir [21], TA recorded values between (0.71 ± 0.01) for the Durona di Cesena and (1.01 ± 0.01 g MA/100 g FW) for the Sweetheart cultivar, while Chockchaisawasdee et al. [20] reported similar TA values for the Pico Negro cultivar (0.6 g MA/100 g FW) and for the Black Star and Puntalazzese cultivars (1.3 g MA/100 g FW).

TPC was expressed as gallic acid equivalents (GAE) per 100 g. TPC recorded differences among cultivars ranging from 76.91 ± 41.83 mg GA/100 g for the Ferrovia cultivar to 132.55 ± 53.19 mg GA/100 g for the Germersdorfer cultivar. The TPC values reported for sweet cherries by Hayaloglu and Demir [21] were similar to those of the present study, ranging between 58.31 ± 10.56 for the Van cultivar to 115.41 ± 7.98 mg GAE/100 g FW for the Belge cultivar. Likewise, Chockchaisawasdee et al. [20] determined TPC values between 50 mg GAE/100 g for the Pico Colarado cultivar to 687.4 mg GAE/100 g for Santina cultivar, while the Ferrovia cultivar recorded higher TPC (97.4 mg GAE/100 g) than those of the present study. Usenik et al. [23] analyzed 13 sweet cherry cultivars from Slovenia and reported a TPC value of (44.3 ± 3.42 mg GAE/100 g FW) for the Lapins cultivar, lower than that of the present study. Finally, Vavoura et al. [11] reported TPC values generally higher than those of the present study, for the four cultivars, Canada Giant, Lapins, Ferrovia, and Skeena, ranging between 95.14 (Canada Giant) and 170.35 (Skeena) mg GAE/100 g FW.

In the penetration test, the Force (Load) showed a wide range of values i.e., 8.00 ± 1.30 N for the Canada Giant cultivar to 25.00 ± 12.10 N for the Germersdorfer cultivar. Esti et al. [22] ran the penetration test for the Ferrovia cultivar and recorded a value of 19.00 ± 0.10 N for Force (Load), similar to that recorded in the present study—17.90 ± 9.00 N. Papapetros et al., [17] reported similar Force (Load) values to those of the present study for cherry cultivars: Kordia, Regina, Mpakirtzeika, and Skeena, ranging between 11.3 and 20.6 N.

As far as sensory evaluation is concerned, all four cultivars showed no differences (*p* > 0.05) for fruit external color. Flesh color differed among cultivars with Lapins showing the darkest and Germersdorfer showing the lightest flesh color. Texture scores showed significant differences (*p* < 0.05) among cultivars tested, with Canada Giant and Germersdorfer showing the most acceptable texture. Likewise, taste differed (*p* < 0.05) for the samples of different cultivars and ranged between 3.89 ± 0.05 for the Ferrovia cultivar to 4.50 ± 0.00 for the Canada Giant cultivar, with the latter considered as having the most acceptable taste. This cultivar also showed the highest TA and given the fact that all cultivars had similar sugar content, it was most probably the balance between sugar/acidity of the Canada Giant cultivar that resulted in the highest preference by panelists. Dever et al. [24] performed sensory evaluation of 16 sweet cherry cultivars originating from three harvest periods (early, midseason, and late harvest). Among them, the Lapins cultivar (midseason harvest) recorded values of 7.2, 6.0, 3.2, 5.8 for the juiciness, sweetness, sourness, and flavor, respectively (10-point scoring scale). Serradilla et al. [7] studied the sourness, sweetness, and fruit flavor for four cherry cultivars. The values for the sourness ranged from 2.56 to 5.13 for Ambrunés and Sweetheart cultivars, respectively (10-point scoring scale). Sweetness recorded larger differences, with the Pico Colorado cultivar showing the lowest and the Ambrunés cultivar the highest value (3.26 and 6.31, respectively). Finally, the fruit flavor ranged among 4.25 for the Pico Negro cultivar to 6.23 for the Ambrunés cultivar. Finally, Vavoura et al. [11] reported significant differences in external and flesh color, texture/firmness, and flavor for the four cultivars, Canada Giant, Lapins, Ferrovia, and Skeena. Differences in sensory parameters between Vavoura et al. and the present study may be related to regional soil and environmental characteristics (Naousa vs. Kozani and Edessa).

### 3.2. Analysis of Sugars

The results for cherry fructose and glucose content are shown in Table 1. More specifically, the Germersdorfer cultivar was found to be the richest in sugars among the four cultivars tested (16.25 ± 3.48 g/100 g for glucose and 4.62 ± 0.97 g/100 g for fructose). On the other hand, the Lapins cultivar had the lowest concentration for both sugars (7.37 ± 1.20 g/100 g glucose and 2.83 ± 0.32 g/100 g fructose). In all samples tested, glucose recorded a substantially higher concentration than that of fructose.

Papapetros et al. [17] reported similar values for glucose and fructose as those of the present study for cherry cultivars: Kordia, Regina, Mpakirtzeika, and Skeena, ranging between 9.2 and 17.9 g/100 g for glucose and between 2.1 and 5.1 g/100 g for fructose. Vursavuş et al. [18] determined the sugars: glucose, fructose, sucrose, and sorbitol in four sweet cherries cultivars, and reported a total amount of sugars equal to 103.87, 108.41, 108.88, and 113.13 g/kg of FW for Larian, Van, Noir de Guben, and 0-900 Ziraat, respectively. Of the sugars determined, glucose was found to have the highest content in cherry samples, followed by fructose, sorbitol, and sucrose. Usenik et al. [23] analyzed 13 cultivars of sweet cherries from Slovenia, including the Lapins cultivar, and reported that its concentration of glucose and fructose was 93.7 ± 3.13 g/kg FW and 79.9 ± 3.40 g/kg FW, respectively. The other tested cultivars recorded glucose content between 61.8 ± 6.67 g/kg FW (for Sylvia cultivar) and 123 ± 4.02 g/kg FW (for the Early Van Compact), while the fructose content ranged from 51.5 ± 5.68 g/kg FW (for the Ferprime cultivar) to 101.5 ± 5.14 g/kg FW (for Lala Star cultivar). Esti et al. [22] determined the conventional quality and sensorial changes in cherries of various cultivars after cool storage and reported high values for sugars, with fructose ranging from 4.8 ± 0.4 to 5.1 ± 0.4 g/100 g and glucose ranging from 5.8 ± 0.4 to 6.4 ± 0.4 g/100 g. In general, the sugar content of sweet cherry cultivars in the present work was of the same order of magnitude to that reported in the literature.

### 3.3. Volatile Compounds

Table 2 presents the groups of volatile compounds identified, including: aldehydes, alcohols, ketones, hydrocarbons and terpenes with aldehydes; ketones and alcohols were the most abundant classes of volatile compounds recorded. The major aldehydes were: acetaldehyde followed by (E)-2-hexenal and hexanal, known as green leaf volatiles and major contributors of cherry fruit flavor [25,26].

The Germersdorfer cultivar had the lowest concentration of aldehydes (0.070 ± 0.019 mg/kg), while the Ferrovia cultivar had the highest concentration (0.214 ± 0.058 mg/kg). Alcohol concentrations ranged from 0.122 ± 0.055 mg/kg in the Germersdorfer cultivar to 0.210 ± 0.102 mg/kg in the Lapins cultivar. From the group of alcohols, ethanol was the compound with the highest concentration in all four cultivars up to 0.113 ± 0.032 mg/kg for the Germersdorfer cultivar. Regarding ketones, the Germersdorfer cultivar recorded the highest concentration, 0.346 ± 0.239 mg/kg, and was the only cultivar in which both acetone and 2-butanone were identified. The last two categories of volatile compounds, i.e., hydrocarbons and terpenes, exhibited very low concentrations in all four cultivars.

Vavoura et al. [11] reported that carbonyl compounds were the most abundant volatile compounds, ranging from 14.75 μg/kg in the Lapins cultivar to 34.62 μg/kg in the Ferrovia cultivar. Alcohols gave the second-strongest signals, ranging from 5.56 for the Ferrovia cultivar to 22.21 μg/kg for the Skeena cultivar. According to Papapetros et al. [17], volatiles decreased in the following order: Skeena > Regina > Mpakirtzeika > Kordia cherry cultivar, with aldehydes being the most abundant class of volatile compounds, followed by alcohols.

Finally, according to Serradilla et al. [7], (E)-2-hexen-1-ol was the main alcohol present in Picato type and Sweetheart sweet cherries in Spain. However, in the present study, this compound was identified only in the Germersdorfer cultivar (0.001 ± 0.001 mg/kg).

### 3.4. Minerals

All four cultivars had similar mineral concentrations. Mineral data are shown in Table 3. The highest amount of minerals was found in the Canada Giant cultivar, followed by the Ferrovia, Lapins, and Germersdorfer cultivars (2662 ± 437, 2452 ± 385, 2278 ± 347, and 2232 ± 351 mg/kg, respectively). The mineral identified with the highest value in the Canada Giant cultivar was Potassium. Phosphorous was the second-most abundant mineral, recording its highest concentration in the same cultivar (282.5 ± 52.1 mg/kg). Calcium and Magnesium both reported high concentrations in the Lapins cultivar (138.3 ± 60.9 and 134.4 ± 38.4 mg/kg, respectively). Minerals such as Be, Cr, Li, Se, Sn, Ti, Tl, and V were also identified, but in a very low concentration, lower than 1 mg/kg.

There are only a few studies in the literature reporting mineral content in cherries. De Souza et al. [27] determined five minerals (P, K, Zn, Mg, Fe) in cherries from Brazil. Potassium was the main mineral (highest concentration equal to 90.92 mg/100 g FW), followed by P and Mg with similar concentrations (12.2 to 12.3 mg/100 g FW), Fe (1.16 mg/100 g FW), and Zn (0.69 mg/100 g FW). It should be noted that Ca was not detected in any of the samples analyzed. The range of concentrations for the main minerals in the above study is similar to those in the present study with the exception of Ca, which was not identified in the above study. Papapetros et al. [17] reported similar mineral content for cherry cultivars: Kordia, Regina, Mpakirtzeika, and Skeena, ranging between 2114 mg/kg for the Kordia cultivar and 2520 mg/kg for the Skeena cultivar. Finally, Matos-Reyes et al. [15] determined minerals in various cherry cultivars grown in Spain. The mineral showing the highest concentration was K ranging from 13,000 mg/kg in samples from Cáceres to 5500 mg/kg in Aragón cherries, followed by Ca and Mg present in concentrations higher than 500 mg/kg. Sodium varied from 10 mg/kg in Huesca sample to 70 mg/kg in Aragón and Castellón samples. The rest of the minerals were present in concentrations lower than 1 mg/kg.

### 3.5. Cultivar Differentiation of Four Cherry Cultivars (Ferrovia, Canada Giant, Lapins, and Germersdorfer) Based on Analytical Parameters

The 56 cherry samples were subjected to MANOVA in order to determine those parameters that are significant for the differentiation of cultivars. Dependent variables initially included the 28 volatile compounds, while cultivar was taken as the independent variable [28]. Pillai’s Trace = 2.619 (*F* = 2.750, *p*-value = 0.001 < 0.05) and Wilks’ Lambda = 0.001 (*F* = 2.763, *p*-value = 0.001 < 0.05) index values showed the existence of a significant multivariable effect of cultivar origin on the identity of cherry volatile compounds. Seven of the 28 volatile compounds were found to be significant (*p* < 0.05) for the differentiation of cherries according to cultivar and thus, were subjected to LDA. In LDA analysis, the cultivar was taken as the dependent variable, while the measured physicochemical parameters were taken as the independent variables [28]. The overall correct classification rate was 77.3% using the original and 65.9% using the cross-validation method, not a very satisfactory rate.

Similar statistical treatment was used for the conventional quality parameters and minerals, which gave a respective correct classification rate of 72.7% and 75%. Sugar statistical analysis showed that only fructose was significant for the differentiation of cultivars and thus, the formation of only one discriminant function showed that the statistical model developed was unable to provide results regarding the differentiation of cherry cultivar. In order to increase the correct classification rate, combinations of analytical sets of data were tested.

The combination of volatile compounds and conventional quality parameters were taken as the dependent variables, while cultivar was taken as the independent variable. Pillai’s Trace = 2.920 (*F* = 4.924, *p*-value = 0.001 < 0.05) and Wilks’ Lambda = 0.001 (*F* = 5.280, *p*-value = 0.001 < 0.05) index values showed that there is a significant multivariate effect of volatile compounds and conventional quality parameters on cherry cultivar. Seven of the volatile compounds and six conventional quality parameters were found to be significant (*p* < 0.05) for the differentiation of cultivar. These were then subjected to LDA. The results of statistical treatment are shown in Table 4. In Figure 1, it is shown that all cultivars are well differentiated, while the Lapins and the Canada Giant are quite close to each other. The overall correct classification rate achieved was 97.7% for the original, while for the cross-validation method, the respective rate was 84.1%, very satisfactory for both methods.

Likewise, the same statistical treatment was applied to the other combinations of sets of analytical data. Statistical analysis of minerals and conventional quality parameters showed that only eight minerals and six of the conventional quality parameters were found to be significant (*p* < 0.05) for cultivar differentiation (Table 4). The overall correct classification rate achieved was 97.7% for the original and 86.4% for the cross-validation method. In Figure 2, it is obvious that Ferrovia and Canada Giant are well differentiated from the other cultivars.

Despite the fact that sugars per se could not provide information on the differentiation of cultivars, their combination with minerals and conventional quality parameters showed that eight minerals, six of the conventional quality parameters, and fructose were found to be significant (*p* < 0.05) for cultivar differentiation (Table 4). The overall correct classification rate achieved was 100% for the original and 88.6% for the cross-validation method, a very satisfactory value for both methods. In Figure 3, it is shown that Ferrovia and Canada Giant are very well differentiated, while Germersdorfer and Lapins are reasonably close to each other. Matos-Reyes et al. [15] determined the mineral content of Spanish cherries from different geographical areas (Aragón, Cáceres, Castellón, Huesca, and Alicante’s Mountain) using ICP-OES. Of the 42 elements determined, only 22 and 23 minerals for stones and fruit edible part were shown to be significant for the differentiation of cherry cultivars, respectively. The classification of cherry stones and edible parts was carried out using LDA. The results showed a correct classification rate of 100% for the edible part of cherries and 96.43% for the stone part. Unfortunately, no mention was made in this study on the cherry cultivars used. Finally, in a similar work, Papapetros et al. [17] reported a correct classification rate equal to 82.1% based on minerals, 89.5% based on conventional parameters, and 89.7% based on volatile compounds for the differentiation of Regina, Kordia, Mpakirtzeika, and Skeena cherry cultivars.

Other combinations tested that gave lower correct classification rates were volatiles and sugars (77.1%), volatiles and minerals (77.3%), conventional quality parameters and sugars (77.5%).

### 3.6. Cultivar Differentiation of All Eight Cherry Cultivars (Ferrovia, Canada Giant, Lapins, Germersdorfer, Kordia, Regina, Skeena, And Mpakirtzeika) Based on Analytical Parameters

Similarly, as described above, all 108 cherry samples were subjected to MANOVA in order to determine those volatile compounds that are significant for the differentiation of cultivars. Nineteen of the thirty volatile compounds were found to be significant (*p* < 0.05) for the differentiation of cherries according to cultivar and thus, were subjected to LDA. The overall correct classification rate was 89.9% using the original and 69.6% using the cross-validation method, not a very satisfactory rate.

Similar statistical treatment was used for the conventional quality parameters and minerals, which gave a respective correct classification rate of 70% and 61.4%. As already stated above, sugar statistical analysis showed that only fructose was significant for the differentiation of cultivars and thus, the formation of only one discriminant function showed that the statistical model developed was unable to provide results regarding the differentiation of cherry cultivar. In order to increase the correct classification rate, combinations of analytical sets of data were tested.

When volatile compounds and conventional quality parameters were combined, the Pillai’s Trace and Wilks’ Lambda index values showed a significant multivariate effect of volatile compounds and conventional quality parameters on cherry cultivar. Nineteen of the volatile compounds and five of the conventional quality parameters were found to be significant (*p* < 0.05) for the differentiation of cultivar. These were then subjected to LDA. Results showed an overall correct classification rate of 98.5% for the original and 85.3% for the cross-validation method, very satisfactory for both methods.

Likewise, the same statistical treatment was applied to the other combinations of sets of analytical data. Statistical analysis of i.e., minerals and conventional quality parameters, showed that only 13 minerals and 5 of the conventional quality parameters were found to be significant (*p* < 0.05) for cultivar differentiation. The overall correct classification rate achieved was 100% for the original and 85.1% for the cross-validation method. In terms of classification rate, the two most successful combinations were: (i) conventional quality parameters plus volatiles plus minerals resulting in an overall correct classification rate of 100% for the original, and 85.5% for the cross validation method, indeed a very satisfactory value for both methods (Table 4, Figure 4a,b); and (ii) the combination: conventional quality parameters, volatiles, minerals, and sugars (fructose), resulting in an overall correct classification rate of 100% for the original and 91.3% for the cross-validation method (Table 4, Figure 5a,b).

In Figure 4a, it is shown that the Skeena and Mpakirtzeika cultivars are very well differentiated from the rest. In Figure 4b, it is shown that the Germersdorfer cultivar is well differentiated from the Regina, Lapins, Ferrovia, and Canada Giant cultivars, but not from the Kordia cultivar. Lapins, Ferrovia, and Canada Giant are considerably overlapping.

In Figure 5a, it is shown that the Skeena and Mpakirtzeika cultivars are again very well differentiated from the rest. In Figure 5b, it is shown that the Canada Giant cultivar is well differentiated from the Ferrovia, Lapins, Regina and Kordia cultivars, but not from the Germersdorfer cultivar. Lapins and Regina as well as Regina and Kordia are considerably overlapping.

## 4. Conclusions

Analysis of volatile compounds, minerals, and conventional quality parameters showed significant differences among cherry cultivars tested. Statistical treatment of the individual sets of data gave acceptable but not satisfactory correct classification rate (volatile compounds: 65.9%, conventional quality parameters: 72.7% and minerals: 75%). Furthermore, combinations of selected data sets increased correct classification rate i.e., volatiles and conventional quality parameters: 84.1%, minerals and conventional quality parameters: 86.4%. Finally, even though sugars per se could not provide information on cherry cultivar differentiation, when combined with minerals and conventional quality parameters, the classification rate was increased to 88.6%. This was the highest rate achieved in the present study, suggesting that the use of multi-element analysis may be a useful tool for cherry cultivar differentiation. In our previous work, Papapetros et al. [17], we achieved a very satisfactory differentiation (97.4%) of the botanical origin of four cherry cultivars (Kordia, Regina, Skeena, and Mpakirtzeika) grown in northern Greece using the same analytical methodology. In the present study, in a similar attempt, we were able, for the first time, to successfully differentiate (classification rate 88.6%) the botanical origin of four additional popular cherry cultivars (Ferrovia, Canada Giant, Lapins, and Germersdorfer) grown in the same greater area and during the same seasons in Greece. Finally, differentiation of all eight cherry cultivars was achieved with a classification rate of 85.5% for the combination of conventional quality parameters, volatiles, and minerals; and 91.3% for the combination of conventional quality parameters, volatiles, minerals, and sugars.

## Figures and Tables

**Figure 1 foods-08-00442-f001:**
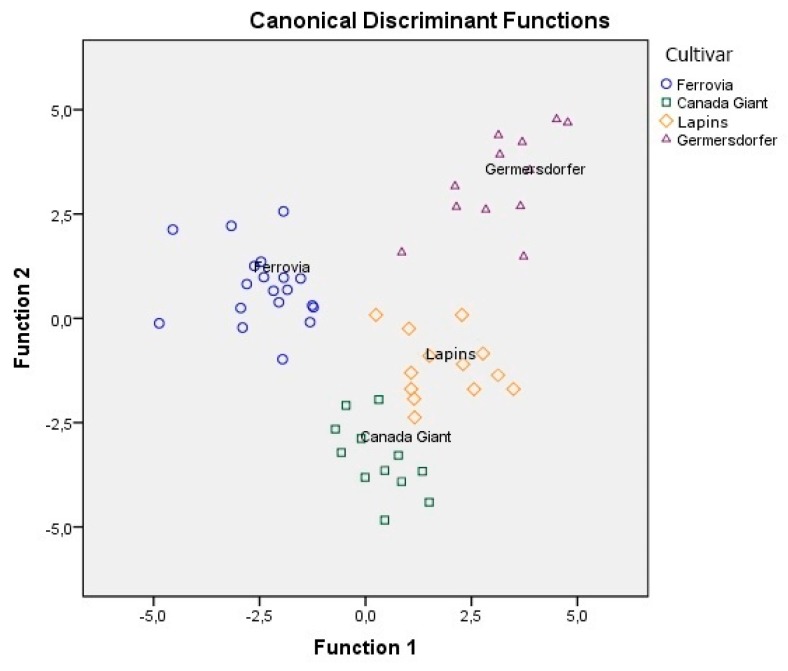
Differentiation of four cultivars based on the combination of volatile compounds and conventional quality parameters.

**Figure 2 foods-08-00442-f002:**
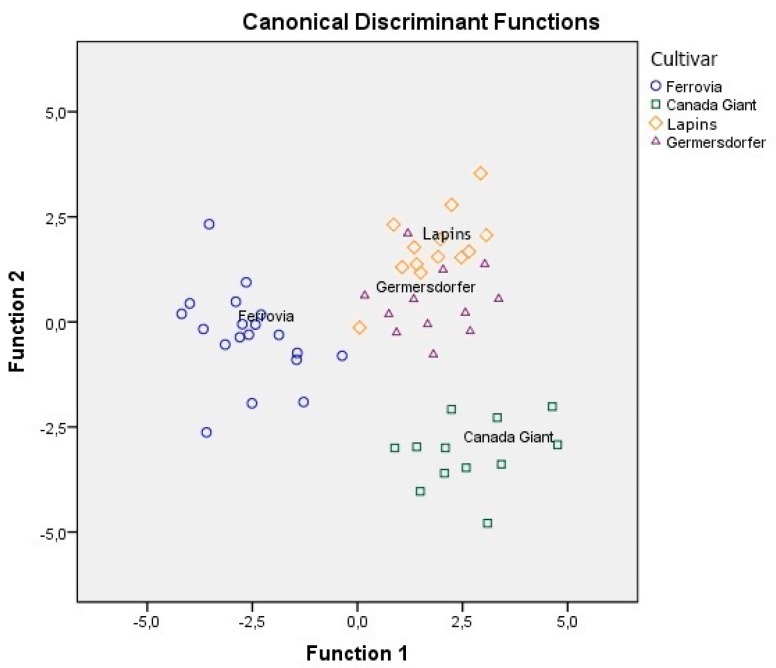
Differentiation of four cultivars based on the combination of minerals and conventional quality parameters.

**Figure 3 foods-08-00442-f003:**
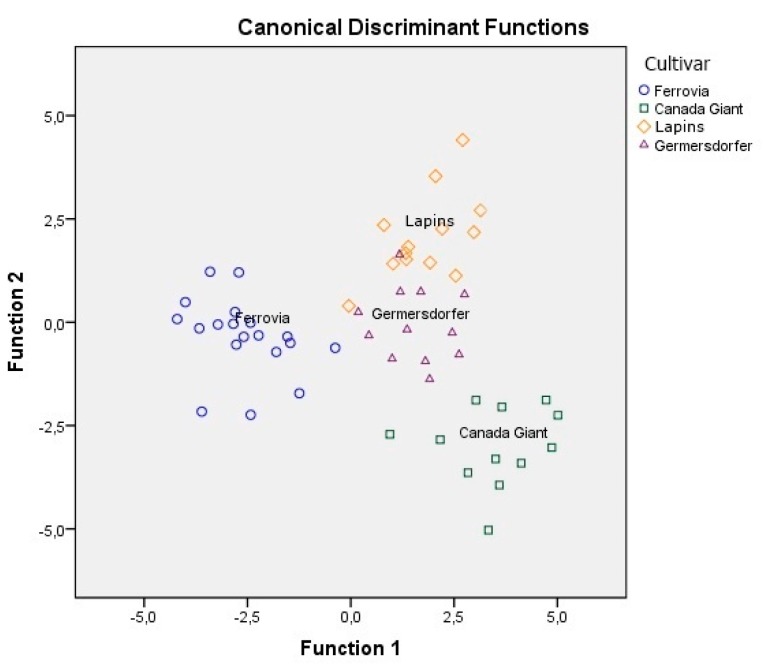
Differentiation of four cultivars based on the combination of minerals, conventional quality parameters, and sugars.

**Figure 4 foods-08-00442-f004:**
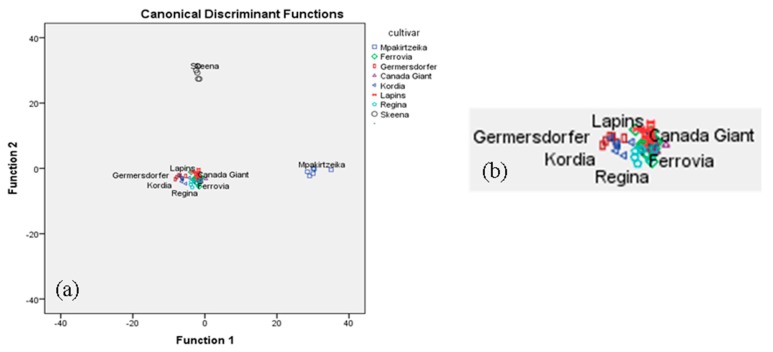
(**a**) Differentiation of eight cultivars based on the combination of conventional quality parameters, volatiles, and minerals; (**b**) blow up of Figure 4a.

**Figure 5 foods-08-00442-f005:**
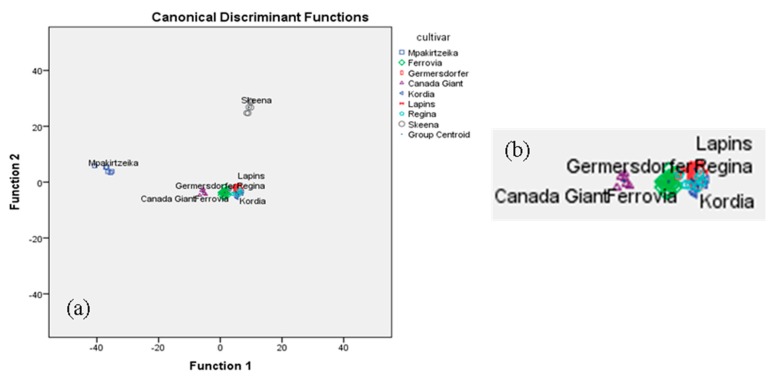
(**a**) Differentiation of eight cultivars based on the combination of conventional quality parameters, volatiles, minerals, and sugars; (**b**) blow up of Figure 5a.

**Table 1 foods-08-00442-t001:** Mean values and standard deviations (SD) of conventional quality parameters and sugars of the cherry samples tested.

Cultivar	Ferrovia	Canada Giant	Lapins	Germersdorfer	*p* *
	Mean ± SD	Mean ± SD	Mean ± SD	Mean ± SD	
**pH**	4.08 ± 0.71 ^a^	4.14 ± 0.09 ^a^	4.05 ± 0.09 ^a^	3.98 ± 0.20 ^a^	0.951
**TSS (Brix)**	14.06 ± 2.27 ^a^	13.11 ± 4.45 ^a^	12.23 ± 1.99 ^a^	12.33 ± 2.33 ^a^	0.219
**TA (gr MAE/100gr FW)**	0.30 ± 0.04 ^ab^	0.35 ± 0.04 ^c^	0.31 ± 0.03 ^b^	0.27 ± 0.04 ^a^	0.001
**TPC (mg GAE/100gr FW)**	76.91 ± 41.83 ^ab^	82.99 ± 8.60 ^a^	128.6 ± 24.2 ^bc^	132.6 ± 53.2 ^c^	0.001
**Fruit diameter (cm)**	2.83 ± 0.23 ^a^	2.58 ± 0.49 ^a^	2.82 ± 0.16 ^a^	2.84 ± 0.00 ^a^	0.107
**External color**	4.87 ± 0.19 ^a^	4.60 ± 0.00 ^a^	4.54 ± 0.52 ^a^	4.60 ± 0.00 ^a^	0.057
**Flesh color**	4.25 ± 0.26 ^ab^	4.30 ± 0.00 ^ab^	4.53 ± 0.51 ^b^	4.00 ± 0.00 ^a^	0.001
**Taste**	3.89 ± 0.05 ^a^	4.50 ± 0.00 ^b^	3.95 ± 0.05 ^a^	4.00 ± 0.00 ^a^	0.001
**Texture**	4.00 ± 0.00 ^a^	4.50 ± 0.00 ^b^	4.00 ± 0.10 ^a^	4.60 ± 0.00 ^c^	0.000
**Force/Load (N)**	17.90 ± 9.00 ^a^	8.00 ± 1.30 ^a^	11.00 ± 2.80 ^a^	25.00 ± 12.10 ^a^	0.055
**Penetration (mm)**	10.37 ± 0.83 ^c^	8.24 ± 0.33 ^a^	9.08 ± 0.65 ^b^	8.73 ± 0.63 ^ab^	0.000
**Glucose (g/100 g)**	12.69 ± 0.08 ^a^	12.01 ± 2.39 ^a^	7.37 ± 1.20 ^a^	16.25 ± 3.48 ^a^	0.173
**Fructose (g/100 g)**	4.23 ± 1.55 ^ab^	3.95 ± 0.62 ^ab^	2.83 ± 0.32 ^a^	4.62 ± 0.97 ^b^	0.030

TSS: Total Soluble Solids; TA: Titratable Acidity; MAE: Malic Acid Equivalents; FW: Fresh Weight; TPC: Total Phenolic Content; GAE: Gallic Acid Equivalents; ^a, b, c^ Mean with different letters in the same row are significantly different; * *p* values are the result of the application of MANOVA (*p* < 0.05 statistically significant).

**Table 2 foods-08-00442-t002:** Mean values and SD (mg/kg) of volatile compounds of cherry samples tested.

Cultivars			Ferrovia	Canada Giant	Lapins	Germersdorfer	*p* ***
Volatiles	RI_exp_ *	RI_lit_ **	Mean ± SD	Mean ± SD	Mean ± SD	Mean ± SD	
**Aldehydes**							
**Acetaldehyde**	<500	<500	0.136 ± 0.045	0.174 ± 0.073	0.126 ± 0.009	0.046 ± 0.034	0.023
**Hexanal**	798	767	0.044 ± 0.013	0.017 ± 0.005	0.028 ± 0.013	0.013 ± 0.005	0.116
**(E)-2-Hexenal**	852	859	0.031 ± 0.012	0.009 ± 0.003	0.026 ± 0.012	0.008 ± 0.003	0.317
**Butanal, 3-methyl-**	614	605	0.003 ± 0.001	n.d.	0.002 ± 0.001	0.003 ± 0.001	0.736
**Total**			0.214 ± 0.058	0.200 ± 0.083	0.182 ± 0.055	0.070 ± 0.019	
**Alcohols**							
**Ethanol**	<500	<500	0.159 ± 0.059	0.203 ± 0.085	0.206 ± 0.120	0.113 ± 0.032	0.176
**3-Buten-1-ol, 3-methyl-**	727	728	0.003 ± 0.001	0.003 ± 0.001	0.004 ± 0.002	0.004 ± 0.002	0.488
**2-Buten-1-ol, 3-methyl**	777	768	0.005 ± 0.002	n.d.	n.d.	0.003 ± 0.001	0.320
**(E)-2-Hexen-1-ol**	862	859	n.d.	n.d.	n.d.	0.002 ± 0.001	0.001
**Total**			0.167 ± 0.078	0.206 ± 0.101	0.210 ± 0.102	0.122 ± 0.055	
**Ketones**							
**2-Butanone**	600	586	n.d.	n.d.	n.d.	0.004 ± 0.001	0.000
**Acetone**	<500	<500	0.128 ± 0.053	0.090 ± 0.045	0.230 ± 0.117	0.342 ± 0.145	0.000
**Total**			0.128 ± 0.091	0.090 ± 0.064	0.230 ± 0.163	0.346 ± 0.239	
**Hydrocarbons**							
**Hexane**	600	600	0.016 ± 0.006	0.018 ± 0.005	0.021 ± 0.008	0.002 ± 0.001	0.310
**Cyclohexane**	657	660	0.006 ± 0.002	0.012 ± 0.003	0.003 ± 0.001	n.d.	0.310
**Heptane**	700	700	0.005 ± 0.003	0.007 ± 0.004	0.009 ± 0.002	0.004 ± 0.001	0.123
**Pentane, 2-methyl-**	559	554	0.001 ± 0.000	0.008 ± 0.002	0.002 ± 0.001	n.d.	0.341
**1,3-Pentadiene**	516	501	0.005 ± 0.002	n.d.	n.d.	0.002 ± 0.001	0.124
**1,3-Butadiene, 2-methyl-**	520	502	0.006 ± 0.003	0.008 ± 0.003	0.001 ± 0.000	0.007 ± 0.002	0.016
**Total**			0.039 ± 0.005	0.053 ± 0.006	0.036 ± 0.008	0.015 ± 0.003	
**Terpenes**							
**dl-Limonene**	1035	1038	0.010 ± 0.006	0.032 ± 0.019	0.001 ± 0.000	0.018 ± 0.004	0.001
**α-Terpinene**	1019	1024	0.001 ± 0.000	0.004 ± 0.002	0.007 ± 0.003	n.d.	0.374
**p-Cymene**	1026	1033	0.015 ± 0.005	0.023 ± 0.006	0.014 ± 0.005	0.003 ± 0.001	0.576
**α-Pinene**	932	940	0.001 ± 0.000	0.008 ± 0.004	0.003 ± 0.001	n.d.	0.475
**α-Thujene**	923	929	0.002 ± 0.001	0.005 ± 0.002	0.002 ± 0.001	n.d.	0.430
**β-Myrcene**	994	986	0.003 ± 0.001	n.d.	n.d.	n.d.	0.530
**1,8-Cineole**	1033	1044	n.d.	n.d.	n.d.	0.001 ± 0.000	0.017
**γ-Terpinene**	1062	1065	0.005 ± 0.002	0.006 ± 0.003	n.d.	n.d.	0.510
**Total**			0.037 ± 0.002	0.078 ± 0.006	0.027 ± 0.002	0.022 ± 0.001	
**Miscellaneous**							
**Acetic acid**	614	605	0.002 ± 0.001	0.001 ± 0.000	0.003 ± 0.001	0.001 ± 0.001	0.813
**Ethyl ether**	<500	<500	0.005 ± 0.002	n.d.	n.d.	0.016 ± 0.005	0.147
**Ethene, 1,1-dichloro-**	540	510	0.017 ± 0.006	0.025 ± 0.008	0.043 ± 0.015	0.016 ± 0.006	0.383
**Chloroform**	622	618	0.016 ± 0.006	0.016 ± 0.006	0.020 ± 0.011	0.013 ± 0.003	0.228
**Total**			0.040 ± 0.008	0.042 ± 0.012	0.066 ± 0.020	0.046 ± 0.007	

* RIexp: experimental retention index, ** RIlit: literature retention index (NIST MS search), *** *p* values are the result of the application of MANOVA (*p* < 0.05 statistically significant), n.d. = not detected.

**Table 3 foods-08-00442-t003:** Mean values and SD of minerals of cherry samples tested.

Cultivar	Ferrovia	Canada Giant	Lapins	Germersodfer	*p* *
	Mean ± SD	Mean ± SD	Mean ± SD	Mean ± SD	
**Al**	0.81 ± 0.50 ^b^	0.59 ± 0.19 ^ab^	1.06 ± 0.56 ^b^	0.43 ± 0.22 ^a^	0.055
**B**	5.22 ± 1.97 ^a^	0.03 ± 0.01 ^a^	4.73 ± 2.06 ^a^	5.32 ± 1.53 ^a^	0.621
**Ba**	0.49 ± 0.17 ^a^	0.03 ± 0.01 ^a^	0.25 ± 0.14 ^a^	0.05 ± 0.01 ^a^	0.770
**Be**	0.02 ± 0.01 ^ab^	n.d.	0.03 ± 0.01 ^b^	n.d.	0.001
**Ca**	130.3 ± 54.5 ^b^	68.63 ± 13.88 ^a^	138.3 ± 60.9 ^b^	88.50 ± 16.00 ^a^	0.018
**Co**	0.01 ± 0.00 ^a^	0.02 ± 0.01 ^a^	0.02 ± 0.01 ^a^	0.01 ± 0.00 ^a^	0.975
**Cr**	0.11 ± 0.08 ^a^	0.09 ± 0.07 ^a^	0.11 ± 0.07 ^a^	0.12 ± 0.04 ^a^	0.858
**Cu**	1.10 ± 0.28 ^b^	1.34 ± 0.25 ^b^	1.19 ± 0.30 ^b^	0.87 ± 0.10 ^a^	0.028
**Fe**	3.05 ± 0.84 ^a^	3.29 ± 0.86 ^a^	3.21 ± 1.79 ^a^	2.34 ± 0.55 ^a^	0.462
**K**	1922 ± 228 ^a^	2186 ± 308 ^b^	1729 ± 308 ^a^	1753 ± 222 ^a^	0.007
**Li**	0.04 ± 0.02 ^a^	0.05 ± 0.01 ^a^	0.02 ± 0.01 ^a^	0.04 ± 0.02 ^a^	0.104
**Mg**	127.4 ± 17.9 ^a^	113.4 ± 23.7 ^a^	134.4 ± 38.4 ^a^	135.6 ± 30.2 ^a^	0.435
**Mn**	1.42 ± 1.01 ^a^	0.94 ± 0.12 ^a^	1.89 ± 0.92 ^a^	0.59 ± 0.23 ^a^	0.138
**Mo**	0.01 ± 0.00 ^ab^	n.d.	0.03 ± 0.01 ^b^	n.d.	0.031
**Ni**	0.06 ± 0.03 ^a^	0.07 ± 0.01 ^a^	0.09 ± 0.03 ^a^	0.10 ± 0.05 ^a^	0.588
**P**	255.5 ± 40.8 ^a^	282.5 ± 52.1 ^a^	260.0 ± 87.0 ^a^	241.4 ± 56.0 ^a^	0.698
**Sb**	0.18 ± 0.07 ^a^	0.21 ± 0.05 ^a^	0.16 ± 0.07 ^a^	0.19 ± 0.04 ^a^	0.444
**Se**	0.01 ± 0.00 ^a^	n.d.	0.01 ± 0.00 ^a^	n.d.	0.452
**Si**	1.99 ± 0.91 ^a^	3.07 ± 4.36 ^a^	1.50 ± 1.20 ^a^	1.72 ± 0.43 ^a^	0.359
**Sn**	0.34 ± 0.17 ^a^	0.35 ± 0.09 ^a^	0.25 ± 0.08 ^a^	0.31 ± 0.04 ^a^	0.918
**Sr**	0.24 ± 0.09 ^b^	0.14 ± 0.06 ^a^	0.33 ± 0.13 ^b^	0.22 ± 0.08 ^ab^	0.011
**Ti**	0.01 ± 0.00 ^ab^	n.d.	0.04 ± 0.05 ^b^	n.d.	0.008
**Tl**	0.47 ± 0.14 ^a^	0.58 ± 0.17 ^a^	0.47 ± 0.24 ^a^	0.44 ± 0.11 ^a^	0.511
**V**	0.03 ± 0.01 ^a^	0.07 ± 0.03 ^b^	0.03 ± 0.01 ^a^	0.10 ± 0.02 ^b^	0.000
**Zn**	1.28 ± 1.05 ^a^	0.79 ± 0.23 ^a^	1.12 ± 0.35 ^a^	0.87 ± 0.35 ^a^	0.673
**Total**	2452 ± 385	2662 ± 437	2278 ± 347	2232 ± 351	

^a, b^ Means with different letters in the same row are significantly different; * *p* values are the result of the application of MANOVA (*p* < 0.05 statistically significant), n.d. = not detected.

**Table 4 foods-08-00442-t004:** Discriminant functions formed and MANOVA results for each function for parameter combinations tested.

Parameters Combinations	Discriminant Function	% of Variance	% Total Variance	Wilks’ Lambda	*X* ^2^	*df*	*p*
**Volatile Compounds-Conventional Quality Parameters (four cultivars)**	1	51.1	51.1	0.012	152.171	39	0.001
	2	33.9	85.0	0.080	87.024	24	0.001
**Minerals—Conventional Quality Parameters (four cultivars)**	1	57.3	57.3	0.014	149.262	36	0.001
	2	25.2	82.5	0.098	81.361	22	0.001
**Minerals—Conventional Quality Parameters—Sugars (four cultivars)**	1	55.4	55.4	0.013	150.892	39	0.001
	2	27.1	82.5	0.088	83.858	24	0.001
**Minerals—Conventional Quality Parameters—Volatile compounds (eight cultivars)**	1	42.4	42.4	0.001	923.667	259	0.001
	2	38.7	81.1	0.001	712.119	216	0.001
	3	6.5	87.6	0.003	504.682	175	0.000
**Minerals—Conventional Quality Parameters—Sugars—Volatile compounds (eight cultivars)**	1	51.9	51.9	0.001	971.743	266	0.001
	2	26.8	78.7	0.001	742.568	222	0.001
	3	10.5	89.2	0.005	542.909	180	0.001

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
