# Peer review of "Physicochemical, Spectroscopic and Chromatographic Analyses in Combination with Chemometrics for the Discrimination of Four Sweet Cherry Cultivars Grown in Northern Greece"

_foods, 2019, doi:10.3390/foods8100442_

Round 1

Reviewer 1 Report

Manuscript Number: foods-523319-R2

Title: Physicochemical, spectroscopic and chromatographic analyses in combination with chemometrics for the discrimination of four sweet cherry cultivars grown in northern Greece Author(s): Spyridon Papapetros, Artemis Louppis, Ioanna Kosma, Stavros Kontakos, Anastasia Badeka, Chara Papastephanou, Michael Kontominas  

Authors have made check and revised the manuscript.Concerning the comments and revision the text in the experimental part now the information added provide a more comprehensive for the reading explanation.

Concerning the comment on sensory properties I need to apologize to the authors, probably I miss the data.

Regarding the dependent and independent variable I continue to be of my opinion from my experience, also in spite to my knowledge and references

-   Brereton R Applied chemometrics for scientist(wiley) 

-   Chemometrics: A Practical Guide by Kenneth R. Beebe, Randy J. Pell, Mary Beth Seasholtz

-   Steven D. Brown, Romá Tauler and Beata Walczak Comprehensive Chemometrics Chemical and Biochemical Data Analysis, 2009 Elsevier

-   InfoMetrix Pirouette Multivariate Data Analysis Software userguide

-   Xlstat 2019 Addinsoft  

In classification LDA what do you want classify? The cultivar, so it is the dependent (categorical) variable 

Perhaps it depends from the interpretation

 From Wikipedia:Statistics synonyms

Depending on the context, an independent variable is sometimes called a "predictor variable", regressor, covariate, "controlled variable", "manipulated variable", "explanatory variable", exposure variable (see reliability theory), "risk factor" (see medical statistics), "feature" (in machine learning and pattern recognition) or "input variable."[9][10] In econometrics, the term "control variable" is usually used instead of "covariate".[11][12][13][14][15]Depending on the context, a dependent variable is sometimes called a "response variable", "regressand", "criterion", "predicted variable", "measured variable", "explained variable", "experimental variable", "responding variable", "outcome variable", "output variable" or "label".[10]"Explanatory variable" is preferred by some authors over "independent variable" when the quantities treated as independent variables may not be statistically independent or independently manipulable by the researcher.[16][17] If the independent variable is referred to as an "explanatory variable" then the term "response variable" is preferred by some authors for the dependent variable.[10][16][17]"Explained variable" is preferred by some authors over "dependent variable" when the quantities treated as "dependent variables" may not be statistically dependent.[18] If the dependent variable is referred to as an "explained variable" then the term "predictor variable" is preferred by some authors for the independent variable.[18]Variables may also be referred to by their form: continuous or categorical, which in turn may be binary/dichotomous, nominal categorical, and ordinal categorical, among others.  [10] Dodge, Y. (2003) The Oxford Dictionary of Statistical Terms, OUP. ISBN 0-19-920613-9 (entry for "regression")[11] Gujarati, Damodar N.; Porter, Dawn C. (2009). "Terminology and Notation". Basic Econometrics (Fifth international ed.). New York: McGraw-Hill. p. 21. ISBN 978-007-127625-2.[12] Introductory Econometrics: A Modern Approach (Fifth ed.). Mason, OH: South-Western Cengage Learning. pp. 22–23. ISBN 978-1-111-53104-1.[13] Last, John M., ed. (2001). A Dictionary of Epidemiology (Fourth ed.). Oxford UP. ISBN 0-19-514168-7.[14] Everitt, B. S. (2002). The Cambridge Dictionary of Statistics (2nd ed.). Cambridge UP. ISBN 0-521-81099-X.[15] Woodworth, P. L. (1987). "Trends in U.K. mean sea level". Marine Geodesy. 11 (1): 57–87. doi:10.1080/15210608709379549.[16] Everitt, B.S. (2002) Cambridge Dictionary of Statistics, CUP. ISBN 0-521-81099-X[17] Dodge, Y. (2003) The Oxford Dictionary of Statistical Terms, OUP. ISBN 0-19-920613-9 

Author Response

Regarding the use of “dependent” and “independent” in the statistical analysis we would like to comment as follows:

As the reviewer noticed in LDA classification the dependent variable is the cultivar. However, in the initial MANOVA analysis the reverse holds; i.e. the reviewer mentions (Wikipedia) “Depending on the context, a dependent variable is sometimes called a "response variable", "regressand", "criterion", "predicted variable", "measured variable", "explained variable", "experimental variable", "responding variable", "outcome variable", "output variable" or "label".

From this statement it is clear that in the MANOVA analysis the dependent variables are the physicochemical parameters measured.

In LDA analysis, however, the reverse holds. (According to Field, 2009, pp 616: “In discriminant analysis we look to see how we can best separate (or discriminate) a set of groups using several predictors (so it is a little like logistic regression but where there are several groups rather than two). It might be confusing to think of actions and thoughts as independent variables (after all, they were dependent variables in the MANOVA!) which is why I refer to them as predictors – this is another example of why it is useful not to refer to variables as independent variables and dependent variables in correlational analysis”) (See revised text, lines 283-285).

In conclusion, we have kept the term “dependent” for the variables measured (volatile compounds, trace elements etc.) in the MANOVA analysis while in the LDA analysis we have used the term “dependent” for the cultivars as the reviewer suggests.

Reviewer 2 Report

The author's reply and modification to the previous manuscript are acceptable.

In particular, I appreciated the addition of the analysis on the data presented in the previous paper merged with the present, which led to a good classification rate of the eight cherry cultivars.

 It extends the validity and robustness of the applied procedure.

Author Response

The reviewer has no comments.